# Health and science-related disinformation on COVID-19: A content analysis of hoaxes identified by fact-checkers in Spain

**Bienvenido León**[1]*, **María-Pilar Martínez-Costa**[1], **Ramón Salaverría**[1], **Ignacio López-Goñi**[2]

**1** Journalism Faculty, Department of Journalism Projects, University of Navarra, Pamplona, Spain,
**2** Medicine Faculty, Department of Microbiology and Parasitology, University of Navarra, Pamplona, Spain

☯ These authors contributed equally to this work.
* bleon@unav.es

**Data Availability Statement:** All content analysis files are available from the Zenodo database. DOI:10.5281/zenodo.4895047 https://zenodo.org/record/4895047#.YLfDrLczbct.

## Abstract

A massive "infodemic" developed in parallel with the global COVID-19 pandemic and contributed to public misinformation at a time when access to quality information was crucial. This research aimed to analyze the science and health-related hoaxes that were spread during the pandemic with the objectives of (1) identifying the characteristics of the form and content of such false information, and the platforms used to spread them, and (2) formulating a typology that can be used to classify the different types of hoaxes according to their connection with scientific information. The study was conducted by analyzing the content of hoaxes which were debunked by the three main fact-checking organizations in Spain in the three months following WHO's announcement of the pandemic (N = 533). The results indicated that science and health content played a prominent role in shaping the spread of these hoaxes during the pandemic. The most common hoaxes on science and health involved information on scientific research or health management, used text, were based on deception, used real sources, were international in scope, and were spread through social networks. Based on the analysis, we proposed a system for classifying science and health-related hoaxes, and identified four types according to their connection to scientific knowledge: "hasty" science, decontextualized science, badly interpreted science, and falsehood without a scientific basis. The rampant propagation and widespread availability of disinformation point to the need to foster media and scientific caution and literacy among the public and increase awareness of the importance of timing and substantiation of scientific research. The results can be useful in improving media literacy to face disinformation, and the typology we formulate can help develop future systems for automated detection of health and science-related hoaxes.

## Introduction

A great amount of misinformation and hoaxes on matters related to the pandemic emerged in parallel with the COVID-19 pandemic, and spread primarily through social networks. This

**Funding:** This study was funded by RRSSalud, via the BBVA Foundation, within the Grants for Scientific Research Teams—Economy and Digital Society, 2019. This study was also funded by IBERIFIER—Iberian Digital Media Research and Fact-Checking Hub, via the European Commission under the call CEF-TC-2020–2 (European Digital Media Observatory), grant number 2020-EU-IA-0252. The grants were not specifically assigned to any author but to the whole project. The funders had no role in study design, data collection and analysis, decision to publish, or preparation of the manuscript.

**Competing interests:** The authors have declared that no competing interests exist.

phenomenon reached such levels that the World Health Organization (WHO) described it as a "massive infodemic," and warned the world of its dangers as it prevents the public from accessing the much-needed reliable information about the disease [1]. It is well known that many of the hoaxes were focused on scientific and health-related topics [2,3]. However, the relationship between scientific information and the characteristics of these hoaxes has not been elucidated.

For the first time in contemporary history, a pandemic of this magnitude was experienced, and all media outlets across the globe disseminated a huge amount of "express science" that gave rise to a problematic relationship between science and society. A lot of information was based on preprints of scientific publications that had not yet undergone a peer-review process, and this contributed to public misinformation.

Spain was hit hard by the pandemic and suffered a high percentage of infections and deaths [4]. On March 14, 2020, the government announced a state of alarm, which involved a nationwide lockdown that lasted until June 21. Spain was restrained by a national lockdown, which created a crisis, and citizens were eager to understand the pandemic better and turned to social media to receive immediate information. Spain was thus a perfect case for our study.

## The phenomenon of information disorder

History is littered with examples of fabrication and dissemination of falsehoods by people, organizations, and governments [5,6]. Recently, public dissemination of falsehoods has reached unprecedented proportions. Digital networks have transformed traditional public communication processes, and one of their consequences is that incorrect information can now be spread worldwide quickly, and on a massive scale.

Disinformation refers to deliberate deception, whereas misinformation refers to the unintended proliferation of falsehoods. These two categories effectively differentiate between acts of malice (voluntary) and mistakes (involuntary). These two broad categories include multiple modalities and specific terms. Research has explored certain modalities such as conspiracy theories [7], rumors [8], and hoaxes [9].

In journalism, "fake" [10,11] or "false" [12] news phenomenon has been widely investigated. Interest in disinformation within the media has intensified over the last decade, especially since 2016, as a result of the US presidential election [13] and the Brexit referendum [14]. The incidence of false information during these events helped popularize the controversial and ambiguous concept of "fake news" [11,15]. According to Tandoc Jr. et al. [16], "fake news" is a multiform reality that encompasses diverse expressions, such as news satire, news parody, fabrication, manipulation, advertising, and propaganda.

Among the different forms and modes of disinformation, hoaxes play a prominent role. These have been defined as "all intentionally false content that appears to be true, conceived with the purpose of deceiving the public and publicly spread via any social platform or social network" [2]. In our study, we opted to use the hoax concept because the falsehoods investigated not only correspond to content disseminated in news media, but according to the extant literature, it is a concept that designates deliberate falsehoods and targets the general public through any communication channel.

Identified as key challenges of our time [17], misinformation and disinformation have been the subject of many research articles with a wide range of approaches and methodologies [18]. The deliberately misleading nature of false information makes it difficult to study and analyze, and most studies conducted thus far have focused on three aspects: (1) identification of the forms of false content, (2) the dynamics of dissemination, especially on social networks, and (3) the impact on public opinion.

A majority of studies on misinformation and disinformation have focused on politics. Research on fake news has also been conducted in the fields of science and health. Waszak et al. [19] studied a sample of news items in the Polish language from 2012 to 2017 and reported that 40% of the content was misinformation, which was extensively shared. In a study on English-language news coverage of the Zika virus, Sommariva et al. [20] found that rumors were shared on social networks three times more often than verified information. A study conducted in the US concluded that misinformation was a worrying issue, particularly due to the easy accessibility and widespread usage of social media [21].

The COVID-19 pandemic has highlighted the seriousness of the problem posed by the spread of health-related falsehoods. According to studies conducted in Spain, during the early stages of the pandemic, the health crisis created a demand for information in addition to traditional media coverage [22], and an emotional need to share relevant information that would help people make decisions about their behavior during a health crisis [23]. This created the perfect breeding ground for the spread of hoaxes.

Social networks provided the public with an alternative means of searching for information about the pandemic, and this was in line with a pattern of behavior that had already been observed in previous crises. A similar change in consumption habits occurred in China during the SARS epidemic in 2003 when the general public actively sought information from alternative sources and created alternative information channels by being information producers and disseminators [24].

Multinational surveys indicated that misinformation about COVID-19 is a global phenomenon [25], and false or misleading information is common, especially in social networks and messaging applications [26].

Misinformation about the pandemic spread rapidly in several countries, although patterns of diffusion among topics vary [27]. Many citizens in several countries viewed misinformation on this topic as highly reliable information, and it negatively affected people's willingness to get vaccinated and comply with guidance measures [28]. Exposure to misinformation about COVID-19 had other relevant effects, including greater information avoidance and less systematic processing of information about the pandemic, as revealed by a multinational study [29].

Other regional studies, such as one conducted in India, revealed that misinformation on COVID-19 increased consistently in 2020, which may be because it was a major event that generated uncertainty and even panic in this country, as in other parts of the world [30].

Regarding the distribution of misinformation, a study conducted in India indicated that online media (especially social media) produced 94.4% of fake news, whereas mainstream media produced only 5.6% [3]. Social media became particularly relevant during the pandemic as it provided an outlet for people's frustrations, and increased their interest at a time when worldwide lockdowns imposed physical constraints on people [31]. Different social media platforms spread different volumes of misinformation because of different interaction patterns and the peculiarity of the audience on each platform [32]. On Twitter, the rate of misinformation was higher among informal individual/group accounts than among other types of accounts [33].

A multinational study on mis- and disinformation about COVID-19 conducted by the Reuters Institute for the Study of Journalism indicated that most of the information disorders published on social media came from "ordinary people," although those pieces of false information which came from politicians and celebrities achieved more engagement [34]. Another study based on Spain suggested that many hoaxes did not include any indicator of "cognitive authority" (e.g. a source of information that is perceived as authorized, based on its knowledge of a topic) [35]. On Twitter and Facebook, a minority of "super spreaders" exerted

a strong influence on each platform, and there was evidence of coordinated sharing misinformation on the pandemic [36]. COVID-19 may have been used as a vector to spread misinformation and disinformation for political purposes [37].

Among the few studies on the actual content of misinformation, research on English-language fact checks found that the bulk of misinformation adopted "various forms of reconfiguration, where true information is spun, twisted, recontextualized, or reworked" [34]. Another study, based in Iran, found that misinformation on COVID-19 disseminated through social media included "disease statistics; treatments, vaccines and medicines; prevention and protection methods; and dietary recommendations and disease transmission ways" [38]. Researchers have also explored conspiracy theories related to COVID-19 (e.g., that it did not exist or was caused by 5G radiation). Evidence suggests that conspiratorial claims are highly unlikely to endure [39].

The current research was based on the results of a previous study on COVID-19 conducted in Spain, which found that one-third of the false information about the pandemic contained falsehoods about scientific and health-related matters and identified four main types of hoaxes: joke, exaggeration, decontextualization, and deception [2].

This study was aimed to analyze the science and health-related hoaxes of COVID-19 that spread during the pandemic. More specifically, our objectives were to (1) identify the characteristics of form and content, and the platforms used to spread science and health-related hoaxes, and (2) formulate a typology that can be used to classify the different types of hoaxes according to their connection with scientific information.

## Materials and methods

### Study design

We used a qualitative method [40]. More specifically, we conducted content analysis, a common methodology in social science, to obtain descriptive indicators through systematic procedures [41]. Our research objectives and methodology were inspired by previous studies that analyzed and classified misinformation models [34] and the hoaxes that were spread during the COVID-19 pandemic [2].

### Data collection

We compiled all hoaxes on COVID-19 published on the websites of Spain's three main fact-checking organizations (Maldita.es, Newtral, and EFE Verifica) over a three months period (March 11 to June 10, 2020), which started when the WHO declared the pandemic.). These are the only Spanish organizations certified by the International Fact-Checking Network (IFCN), which was created in the United States in 2015 by the Poynter Institute, an entity that assesses the quality of the work of fact-checking organizations worldwide [42].

The sample was manually selected by checking the websites of the three organizations and compiling all the items related to COVID-19. Automated or specific data extraction tools were not used. After removing the hoaxes which were repeated on one or more of these three websites, we set up a database with our sample (N = 533), which consisted of hoaxes identified by Maldita.es (N = 327), Newtral (N = 143), and EFE Verifica (N = 63).

### Coding

After classifying the hoaxes in a database, a codebook was developed based on a previous study [2]. After discussing this code in two research team meetings, two independent coders carried out a pre-test in which they jointly coded 5% of the hoaxes to detect compression problems

**Table 1. List of variables.**

| |
|---|
| 1. Subject of the hoax: science/health, politics/government, other.<br>Next, all science/health hoaxes (N = 187) were coded, and variables 2 to 10 were analyzed. |
| 2. Platform used to spread the hoax: networks (in general), Twitter, Facebook, WhatsApp, Instagram, YouTube, and others. |
| 3. Formats used: text, audio, image, video, other. |
| 4. Geographical scope: local, national, international, unspecified/not applicable. |
| 5. Type of hoax: joke, exaggeration, decontextualization, deception. |
| 6. Topic of hoaxes related to science/health: scientific research, scientific policy and health management, advice issued to the public, and others. |
| 7. Topic of hoaxes related to scientific research: origin of the virus, transmissibility, fatality rate, treatments, vaccines, etc. |
| 8. Source type: anonymous, spoofed, fictitious, real. |
| 9. Non-anonymous sources: members of the public, business, government, professional, healthcare/science. |
| 10. Type of healthcare/science sources: researchers, international scientific organizations, national scientific organizations, health professionals, and others. |

and unify the coding criteria. In the testing phase, the final analysis code was obtained, which included the variables detailed in Table 1 (the coding criteria are explained in detail in the codebook attached as Supporting Information, S1 Appendix: Codebook).

Once coding was conducted based on the final codebook, an intercoder reliability test was conducted. It consisted of the double blind codification of 126 items, a representative sample of the initial universe of 187 items (50% of heterogeneity, 5% error margin, 95% level of confidence).

The agreement between the two independent coders was tested through a Cohen's kappa test for the parallel and blind codification of each variable, resulting in an optimal level of agreement for all the variables: Topic 1, 0,96; Topic 2, 0.95; Topic 3, 0.97; Source 1, 0.98; Source 2, 0.95; Source 3, 0.99; Geographical scope, 1; Type of hoax, 0.97 (see S2 Appendix: Statistical analysis).

A chi-squared test was used to analyze the relationship between two categorical variables to contrast the significance of the relationship between them. The results are indicated in the paper where appropriate, and are included in detail in the S2 Appendix: Statistical Analysis Appendix.

## Results

### Content and platforms

The hoaxes were initially subclassified into three main categories: "science and health," "politics/government," and "other." (See Table 2). The primary interest was in identifying hoaxes related to science and health in the context of other social, political, and governmental issues.

**Table 2. Hoax topics.**

| | n | %(*) |
|---|---|---|
| Science and Health | 187 | 35.08 |
| Politics | 176 | 33.02 |
| Other | 170 | 31.89 |
| Total | 533 | 100.00 |

* Percentages are rounded.

**Table 3. Content of hoaxes in the "other" category.**

|  | n | % (*) |
|---|---|---|
| Scams | 58 | 34.12 |
| Public safety | 29 | 17.06 |
| Shocking events | 27 | 15.88 |
| People's behavior | 21 | 12.35 |
| Demonstrations | 19 | 11.18 |
| Celebrities | 8 | 4.71 |
| Predictions | 5 | 2.94 |
| Other (**) | 3 | 1.76 |
| Total | 170 | 100.00 |

* Percentages are rounded.

** These hoaxes concerned insurance companies, employment, or pollution.

This category included several subcategories, such as "scientific policy and health management," "scientific research," and "advice issued to the public" related to the COVID-19 pandemic. The "politics" category included content that focused on political parties, party members, and government affairs at international, national, regional, or local level.

A relevant share of the hoaxes fell into the "other" category (see Table 3). Among them, scams were the most prominent. These included fake job offers, fraudulent requests for personal data, the sale or promotion of nonexistent products, and announcements of vouchers and aid for buying food and medicine. Second, hoaxes were also related to public safety. These included actions by security forces (e.g., municipal, regional, and national police, civil guards, and military emergency units) as well as control of communications and state roads, among others.

The highest number of hoaxes was observed during the first month of study. Of these, just over one-third corresponded to "science and health" hoaxes (see Table 4). The distribution of the content of hoaxes by month was not statistically significant (chi-square = 2,413; df = 4 y p-tail = 0,660), which means there was not a specific month in which a significantly higher proportion of hoaxes about unk > "science and health" was spread.

A total of 187 hoaxes about "science and health" were spread through various platforms, including social networks, the media, and other channels such as SMS, emails, blogs, and non-journalistic websites. Certain hoaxes were found to be spread on more than one platform. This produced a total number of frequencies (N = 218) that was greater than that of the total number of hoaxes in the initial "science and health" sample (N = 187).

Messaging and social media applications (both specified and unspecified) were the most frequently used channels for spreading hoaxes. WhatsApp was the platform used to spread the

**Table 4. General content of hoaxes by month.**

| Date | Science and Health | Politics | Other | Total | |
|---|---|---|---|---|---|
|  |  |  |  | n | % (*) |
| Month 1 (March 11 to April 10) | 109 | 91 | 93 | 293 | 54.97 |
| Month 2 (April 11 to May 10) | 36 | 37 | 30 | 103 | 19.32 |
| Month 3 (May 11 to June 10) | 42 | 48 | 47 | 137 | 25.70 |
| Total | 187 | 176 | 170 | 533 | 100.00 |

*Percentages are rounded.

**Table 5. Platforms used to spread "science and health" hoaxes.**

|  | n | % (*) |
|---|---|---|
| Social media (unspecified) | 72 | 33.03 |
| WhatsApp | 54 | 24.77 |
| Twitter | 26 | 11.93 |
| Facebook | 18 | 8.26 |
| Media outlets | 12 | 5.50 |
| YouTube | 12 | 5.50 |
| Instagram | 4 | 1.83 |
| Other | 20 | 9.17 |
| Total | 218 | 100.00 |

*Percentages are rounded.

maximum number of hoaxes, followed by social networks like Twitter and Facebook, the video-sharing platform YouTube, media outlets, and other platforms (see Table 5).

## Content of science and health-related hoaxes

The frequencies of the three main topics were relatively similar. The most frequent hoaxes were related to scientific research, which will be further analyzed in detail in the study. This was followed by hoaxes related to scientific policy or health management, which focused on issues such as decisions by the authorities to control the spread of the virus and management of health resources. Erroneous advice on how to avoid the coronavirus was also common and included recommendations such as drinking different beverages, following diets, and engaging in practices such as gargling, inhaling steam, and consuming chlorine dioxide (see Table 6).

Most hoaxes on scientific research concerned the origin of the coronavirus; for example, they propagated that the virus was manufactured and released by China or the United States, or connected it to 5G technology. Hoaxes that recommended bogus treatments and vaccines were also common. Hoaxes related to the fatality rate and transmissibility of the virus were also detected, although to a lesser extent (see Table 7).

In some cases, the origin of hoaxes was related to studies that were in their initial stages and had not yet produced any conclusive results. For example, social networks were flooded with images of skin lesions caused by the coronavirus, which caused an alarm among the public [43]. Such lesions were not caused directly by the virus and were, therefore, not contagious, but rather a manifestation of the body's immune response to the virus, or were minor skin lesions associated with inflammation of the skin's blood vessels, a phenomenon that is still being researched.

**Table 6. Content of science and health-related hoaxes.**

|  | n | % (*) |
|---|---|---|
| Scientific research | 64 | 34.22 |
| Scientific policy and health management | 63 | 33.69 |
| Erroneous advice issued to the public | 51 | 27.27 |
| Other | 9 | 4.81 |
| Total | 187 | 100.00 |

* Percentages are rounded.

**Table 7. Content of hoaxes relating to scientific research.**

|  | n | % (*) |
|---|---|---|
| Origin of the virus | 27 | 42.19 |
| Treatments | 16 | 25.00 |
| Vaccines | 10 | 15.63 |
| Fatality rate | 3 | 4.69 |
| Transmissibility | 3 | 4.69 |
| Other | 5 | 7.81 |
| Total | 64 | 100.00 |

* Percentages are rounded.

Based on preliminary scientific results, it was claimed that smoking protected against coronaviruses, a statement that went viral on Twitter [44]. This idea arose from an article published as a preprint that suggested that the percentage of smokers among hospitalized COVID-19 patients in China was lower than that of the general population, which suggested a certain protective role. Subsequently, another preliminary, unreviewed article written by a group of French scientists suggested that nicotine could have preventive and healing effects against COVID-19. The researchers themselves warned people against jumping to premature conclusions and urged them to make a distinction between nicotine and tobacco, a product that contains thousands of toxic substances. This suggests that nicotine could be an effective treatment for acute COVID-19 in a clinical setting, and does not equate to the claim that tobacco can prevent the disease. However, this article was criticized for its serious methodological shortcomings.

Other hoaxes arose from poorly interpreted preliminary research. For example, the claim that COVID-19 was not being treated properly because it was not pneumonia but thrombosis went viral on social media [45]. This claim was based on the results of the first autopsies performed in Italy, in which disseminated intravascular coagulation was detected. Nevertheless, thrombosis was not a cause of COVID-19, but was a consequence of some of more severe cases.

## Science and health-related hoaxes, geographical scope, and format

The most common type of hoax was deception, in which entirely false content was communicated and made credible through various mechanisms. These were followed by hoaxes based on decontextualization, in which the information was placed in a false context. This includes several types of incorrect attributions, such as geographical or chronological displacement. For example, one of the debunked hoaxes stated that during the pandemic, the Spanish prime minister would have a personal medical team of 14 people as a special measure against the coronavirus [46]. In fact, this was a regular medical team assisting the prime minister, not one specially created to protect him during the pandemic. Exaggerations, which accounted for 13.90% of the total, were hoaxes in which facts were represented disproportionately. For example, a headline in a digital publication claimed that singer Shakira had been "hit by coronavirus," thereby implying that she had contracted the disease. However, the body of the article revealed that she had been suffering because her parents lived in a high-risk city during the pandemic [47], and had not been infected herself.

The few parodies detected in our sample were hoaxes intended as mockery or had satirical purposes. For example, an advertisement from a person who was offering to infect people with the coronavirus for a modest price of 60 euros was spread on social media [48] (Table 8).

**Table 8. Types of hoaxes.**

|  | n | % (*) |
|---|---|---|
| Deception | 116 | 62.03 |
| Decontextualization | 43 | 22.99 |
| Exaggeration | 26 | 13.90 |
| Parody | 2 | 1.07 |
| Total | 187 | 100.00 |

* Percentages are rounded.

There were no major differences between the formats used for the different types of hoaxes; the most common was text, followed by photos. Videos and audio clips were less frequent. The only significant difference was that audio clips were used more frequently for exaggerations (chi-square = 45,494**; df = 30; p-tail = 0,035) (see Table 9).

International hoaxes were the most common among all types, apart from parodies. This was followed by hoaxes at the national level and those at the local level. Deception (+), decontextualization (+), and exaggeration occurred more frequently at a global level, than at regional or national level. However, decontextualization and exaggeration occurred more commonly at a national level, than at local level. In contrast, deception occurred more frequently at the local level than at a national level (see Table 10). These differences were not statistically significant (chi-square = 7,256; df = 9; p-tail = 0,6610) and, therefore, none of the types of hoaxes was significantly more prominent in a specific geographical scope.

## Sources

Our study identified four types of sources for these hoaxes: real, anonymous, spoofed, and fictitious (see Table 11). Real sources were individuals and corporations accurately identified; anonymous sources were those that did not identify themselves; and spoofed sources were those sources to which information was falsely attributed.

51.90% of the three non-anonymous types of sources were scientists and health professionals. This category was dominated by health professionals (44.12%), researchers (29.41%), and international scientific organizations (17.65%).

The classification of non-anonymous sources according to the type of hoax (Table 12) revealed several significant relationships (chi-square = 27,090*; df = 18; p-tail = 0,077). Healthcare and scientific sources dominated three out of the four types of hoaxes (deception,

**Table 9. Types of hoaxes, according to format (**).**

|  | Deception | | Decontextualization | | Exaggeration | | Joke | | Total | |
|---|---|---|---|---|---|---|---|---|---|---|
|  | n | % | n | % | n | % | n | % | n | % (*) |
| Text | 66 | 51.56 | 31 | 56.36 | 19 | 54.29 | 2 | 50.00 | 118 | 53.15 |
| Photo | 27 | 21.09 | 15 | 27.27 | 6 | 17.14 | 2 | 50.00 | 50 | 22.52 |
| Video | 26 | 20.31 | 9 | 16.36 | 3 | 8.57 | 0 | 0.00 | 38 | 17.12 |
| Audio | 9 | 7.03 | 0 | 0.00 | 7 | 20.00 | 0 | 0.00 | 16 | 7.21 |
| Total | 128 | 100.00 | 55 | 100.00 | 35 | 100.00 | 4 | 100.00 | 222 | 100.00 |

* Percentages are rounded.

** Some hoaxes used more than one format simultaneously, so the total number of cases in this table (N = 222) was higher than the number of science and health-related hoaxes analyzed (N = 187).

**Table 10. Types of hoaxes according to geographical scope.**

|  | Deception | | Decontextualization | | Exaggeration | | Joke | | Total | |
|---|---|---|---|---|---|---|---|---|---|---|
|  | n | % (*) | n | % (*) | n | % (*) | n | % (*) | n | % (*) |
| International | 57 | 49.14 | 18 | 41.86 | 10 | 38.46 | 0 | 0.00 | 85 | 45.45 |
| National | 23 | 19.83 | 14 | 32.56 | 9 | 34.62 | 1 | 50.00 | 47 | 25.13 |
| Local | 27 | 23.28 | 10 | 23.26 | 5 | 19.23 | 1 | 50.00 | 43 | 22.99 |
| Not specified/ not applicable | 9 | 7.76 | 1 | 2.33 | 2 | 7.69 | 0 | 0.00 | 12 | 6.42 |
| Total | 116 | 100.00 | 43 | 100.00 | 26 | 100.00 | 2 | 100.00 | 187 | 100.00 |

\* Percentages are rounded.

decontextualization, exaggeration), ahead of government sources, and most of these hoaxes were based on deception. For example, a hoax published on May 7, 2020 [49] claimed that coffee consumption prevented and cured the coronavirus, and falsely attributed the claim to Chinese ophthalmologist Li Wenliang, who warned about the coronavirus outbreak and ended up dying from the illness.

Healthcare/scientific sources accounted for a prominent share of hoaxes based on decontextualization; a fine example would be a particular hoax published on March 20, 2020 [50], which included false information about the effects of the coronavirus, and falsely attributed those claims to Spanish Doctor Quique Caubet. In this case, Dr. Caubet acknowledged that he had shared the message through social networks, although he himself had not written it. Government sources were more often quoted in decontextualization hoaxes than in deception hoaxes.

Healthcare/scientific sources contributed mainly to exaggeration hoaxes, ahead of businesses and members of the public. For example, a disproved claim by Newtral on March 25, 2020 [51] stated that sunbathing for half an hour a day boosted immunity against the virus. This misinformation was attributed to two researchers from the University of Turin, who stated that according to preliminary data from a study, it might be useful to recommend that people expose themselves to sunlight as much as possible. They, however, never claimed that sunbathing could prevent infection. Finally, only one joke or parody hoax was detected from a non-anonymous source (professional).

## Discussion

### General content and platforms

The increased demand for information about COVID-19 led to the spread of hoaxes on an extensive variety of subjects, including the health situation, research on the new virus, political

**Table 11. Types of sources.**

|  | n | % (*) |
|---|---|---|
| Real | 78 | 41.71 |
| Anonymous | 56 | 29.95 |
| Spoofed | 45 | 24.06 |
| Fictitious | 8 | 4.28 |
| Total | 187 | 100.00 |

\* Percentages are rounded.

**Table 12. Non-anonymous source types by hoax type.**

|  | Deception | | Decontextualization | | Exaggeration | | Joke | |
|---|---|---|---|---|---|---|---|---|
|  | **n** | **% (*)** | **n** | **% (*)** | **n** | **% (*)** | **n** | **% (*)** |
| Healthcare/science | 48 | 57.83 | 11 | 40.74 | 11 | 55.00 | 0 | 0.00 |
| Government | 9 | 10.84 | 8 | 29.63 | 0 | 0.00 | 0 | 0.00 |
| Member of the public | 8 | 9.64 | 3 | 11.11 | 4 | 20.00 | 0 | 0.00 |
| Business | 5 | 6.02 | 1 | 3.70 | 4 | 20.00 | 0 | 0.00 |
| Professional | 7 | 8.43 | 2 | 7.41 | 0 | 0.00 | 1 | 100.0 |
| Other | 6 | 7.23 | 2 | 7.41 | 1 | 5.00 | 0 | 0.00 |
| Total | 83 | 100.00 | 27 | 100.00 | 20 | 100.00 | 1 | 100.00 |

* Percentages are rounded.

management, and social behavior in response to the crisis. The data from our study indicated that the hoaxes with scientific and health-related content accounted for a considerable percentage (35.08%) of all false information spread during the first three months of the pandemic. Almost half of all hoaxes (43.7%), including 54.9% of science and health-related hoaxes, were spread in the first month of the pandemic. This suggests that in the early days, the strong public interest in accessing information to adapt to a novel situation, combined with a lack of information, gave rise to numerous hoaxes. However, hoaxes about science and health continued to spread consistently during the three sampling months, which makes it difficult to establish a causal relationship between the diffusion of misinformation and specific events beyond the beginning of the pandemic that coincided with the first few weeks of lockdown in Spain.

With respect to platforms, social networks, including both private messaging applications and open networks, provided the primary setting for the spread of hoaxes (82.9%), way ahead of the traditional media and other interpersonal communication channels. The results of this study are affected by our sample selection method, which limited the sample to hoaxes debunked by the three fact-checking organizations. Our results were consistent with previous reports that indicated that the usage rate of social networks increased during times of crisis because social media provided a platform for emotional communication, and were timely and uncensored by government sources [52].

This increased use of social networks went hand-in-hand with an increase in the spread of hoaxes. One of the first studies on this subject carried out in the context of the pandemic indicated that 88% of false information originated on social media platforms [34]. In fact, Spain was one of the European countries where the use of social networks increased the most during the lockdown [53] and where social media acted as the main channel for spreading hoaxes, as reflected in this research.

## Content of science and health-related hoaxes

The large amount of scientific information hastily produced in the first few months of the pandemic created serious communication problems. Many scientific articles were published in such a short time that scientists themselves and even specialized publishers were unable to process them properly.

As mentioned, hoaxes concerning the origin of the coronavirus were among the most common of all science and health-related hoaxes. One case related to this subject was paradigmatic with respect to the way in which some hoaxes were created. One of the main sources of misinformation about the origin of the coronavirus was probably an article published in a preprint format that suggested that the new SARS-CoV-2 virus was a manufactured combination of

HIV and SARS viruses [54]. This preliminary article was withdrawn by the authors within three days of being published after errors were discovered in their bioinformatics analysis and interpretation. However, it was one of the most talked about hoaxes on social networks at the time and promoted the false notion that SARS-CoV-2 was genetically engineered in a laboratory.

On other occasions, disinformation was generated when politics and scientific fraud came together. One of the most prominent cases is the hydroxychloroquine treatment scandal. Preliminary studies had shown that this compound inhibits viral replication in vitro [55]. These findings indicated that hydroxychloroquine was one of the first antivirals to be tested in severe COVID-19 cases. French microbiologist Didier Raoult, advisor to the French government in the fight against the pandemic, promptly claimed that this compound was an effective treatment for COVID-19 in humans. The WHO even included the medication in its international clinical trial. The issue was further clouded by US President Donald Trump's revelation that he was taking hydroxychloroquine to protect himself from coronavirus. To complicate matters further, an article published in *The Lancet* warned that hydroxychloroquine was not only useless, but also associated with adverse effects and an increased risk of death [56]. However, the study did not follow experimental protocols and a group of scientists questioned its results. The work could not be independently verified; therefore, *The Lancet* was forced to retract the article. The efficacy of hydroxychloroquine became a political issue, with some "in favor" and others "against" the treatment, based primarily on ideological rather than scientific reasons.

## Types of hoaxes, geographical scope, and format

Science and health-related hoaxes about COVID-19 took all the forms identified in a previous study [2]. The most frequent were deceptions based on simple untruths. Less frequent, although still prominent, were decontextualization and exaggeration hoaxes, which were created based on partially true information that was then distorted to create the hoax.

Decontextualization appears to be a misinformation strategy that is more frequent in health and science-related hoaxes (22.99%), which includes hoaxes on any topic (16.1%). However, our results do not allow us to identify distinguishing characteristics among decontextualization hoaxes in science and health compared to other types of hoaxes, as there are no significant differences in frequencies regarding format, geographical scope, or sources (except for a small difference in governmental sources, which are more frequent).

The most common formats for these hoaxes were those that required the least technical expertise. Text was the most common format, with photos, videos, and audio clips trailing behind. These data seem to indicate that any member of the public could create hoaxes, and that no special technical skills were required.

In addition, it seems that some formats are especially effective for certain types of hoaxes. For example, decontextualization frequently used photos, and exaggerations were particularly suited to audio clips. However, in our study, the format frequency varied slightly according to the type of hoax.

The fact that many of the science and health-related hoaxes were international in scope was consistent with the global nature of the COVID-19 pandemic, which prompted the public to constantly search for information in other countries, either to compare different pandemic situations or to seek valid reference points to cope with the crisis.

## Sources

Generally, the credibility of hoaxes was based on the source's epistemological authority (whether alleged or real). In the four types of sources that were identified (real, anonymous,

**Table 13. Most common characteristics of hoaxes about science and health.**

| Topics | Scientific research (origin of the virus) and health management |
|---|---|
| Platform | Social networks |
| Format | Text |
| Type of hoax | Deception |
| Scope | International |
| Source | Real |

spoofed, and fictitious), the source was usually presented as a person or institution with a good reputation and/or competence to deal with the subject in question. It was unsurprising that more than half of the sources of science and health-related hoaxes identified (51.90%) were scientists or health professionals.

Sometimes the authority of health professionals was underpinned by their status as "eyewitnesses" (whether alleged or real) to the events in question. Meanwhile, the authority of scientists (international or national researchers and scientific organizations) was based on their reputation and competence. The use of such sources made the hoax seem credible because in areas where science plays a key role, the public generally trusts scientists above friends and family as the main sources of information [57].

The classification of sources by hoax type indicates that different types of sources were used to spread different types of hoaxes. This suggests that all types of hoaxes were underpinned by similar arguments to reinforce the epistemological authority of the sources regardless of the type of source in question.

## Common characteristics

Based on the study's results and the diverse nature of the hoaxes, we can highlight some of the most common characteristics of science and health-related hoaxes spread during the COVID-19 pandemic (Table 13).

## Classification of hoaxes

The results of our study allowed us to identify four types of hoaxes, according to their relationship with scientific knowledge: "hasty" science, decontextualized science, badly interpreted science, and falsehood without a scientific basis. Table 14 lists the typical characteristics of each of these types in terms of origin, source, hoax type, and common topics.

## Limitations

The main limitation of our study is that it analyzed only those hoaxes identified by three fact-checking organizations in a single country (Spain) spread only during the first three months of

**Table 14. Classification of science and health-related hoaxes according to their connection to scientific knowledge.**

| | Origin | Sources | Type of hoax | Common topics |
|---|---|---|---|---|
| Hasty science | Provisional results | Real | Decontextualization, exaggeration | Scientific research (origin of the virus, treatments, vaccines) |
| Decontextualized science | Provisional or definitive results | Real | Decontextualization | Scientific research, scientific policy / health management |
| Badly interpreted science | Definitive results | Real, spoofed | Exaggeration | Scientific research, erroneous advice issued to the public |
| Falsehood without scientific basis | Unknown | Spoofed, anonymous | Deception | Scientific policy / health management, erroneous advice issued to the public |

the pandemic. The fact checkers followed several methods to identify the hoaxes. In some cases, they checked the information sent directly to them from the audience. In other cases, they selected information previously published by news media. This method has its limitations, as fact-checkers cannot address all the misinformation that is spread over a particular period and may have selection biases [58], and such biases could have transitively had an impact on the accuracy of our results. Another limitation is that the data we obtained cannot be easily compared to those of other studies because of the different criteria used for classification.

Our study also suggests that preprints and decontextualization may play a relevant role in creating hoaxes. A profound analysis of these relationships surpasses the aim and scope of our research and our sampling method does not allow the establishment of hoaxes originating from a preprint nor a more profound analysis of the differential characteristics of hoaxes based on decontextualization.

## Conclusion

Our first objective was to analyze the characteristics of the form and content of hoaxes and the platforms used to spread them. Based on the results and discussion presented, we can state that scientific knowledge (whether rigorous or not) played a very prominent role in shaping the hoaxes related to COVID-19. A broad, diverse variety of hoaxes based on scientific knowledge was created and contributed substantially to public misinformation during the pandemic.

Our findings allowed us to identify several characteristics of hoaxes, providing relevant information that can be used as a basis for future research. They can also contribute to a better understanding of how disinformation is spread to the public and, therefore, can help improve media literacy actions to address disinformation about health and science.

Our second objective was to formulate a typology for science and health-related hoaxes. We identified four types according to their connection to scientific knowledge: hasty science, decontextualized science, badly interpreted science, and falsehood without a scientific basis. This typology can serve as a preliminary framework for future research and can help develop systems for automated detection of health and science-related hoaxes.

Many hoaxes (hasty, decontextualized, and poorly interpreted science) have an actual scientific basis, with varying degrees of rigor. In these cases, real or spoofed sources were often used to give the hoax epistemological authority and make it more credible. Sometimes, the origin was a low-quality scientific publication, either in preprint format or in a peer-reviewed article in a prestigious journal.

Decontextualization played a relevant role in health and science-related hoaxes, probably because they were easy to produce, and at the same time, were credible since they were partially true.

Our findings also suggested that preprints might be a prominent source of hoaxes. The spread of hoaxes based on provisional results underlines that scientific research requires time and substantiation, both of which are not feasible during the pandemic. This lack of time to develop comprehensive scientific research was instrumental in contributing to public misinformation.

Certain hoaxes were mere falsehoods without any scientific basis, frequently supported by spoofed or anonymous sources. Theoretically, the public should be able to detect this type of hoax more easily than hoaxes based on actual scientific knowledge. However, this requires a certain level of media and scientific literacy that not all members of the public possess.

The connection between preprints and hoaxes, as well as a deeper analysis of the differential characteristics of hoaxes based on decontextualization, present a desirable topic for future research.

## Supporting information

**S1 Appendix. Codebook.** Coding criteria and variables developed to classify the hoaxes. (DOCX)

**S2 Appendix. Statistical analysis.** Cohen's kappa test was used for inter-coder reliability and the chi-square test was used for significance between variables. **Coding dataset.** Accessible on Zenodo. DOI:10.5281/zenodo.4895047. https://zenodo.org/record/4895047#.YLfDrLczbct. (DOCX)

## Acknowledgments

We would like to thank Editage (www.editage.com) for English language editing.

## Author Contributions

**Conceptualization:** Bienvenido León, Ramón Salaverría, Ignacio López-Goñi.

**Formal analysis:** Bienvenido León, María-Pilar Martínez-Costa.

**Methodology:** Ramón Salaverría.

**Writing – original draft:** Bienvenido León, María-Pilar Martínez-Costa, Ramón Salaverría, Ignacio López-Goñi.

**Writing – review & editing:** Bienvenido León, María-Pilar Martínez-Costa.

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
