## [Decision Letter · Decision Letter 0]

25 Mar 2021

PONE-D-20-35335

Health and science-related misinformation on COVID-19. A content analysis of hoaxes identified by fact checkers in Spain

PLOS ONE

Dear Dr. León,

Thank you for submitting your manuscript to PLOS ONE. After careful consideration, we feel that it has merit but does not fully meet PLOS ONE’s publication criteria as it currently stands. Therefore, we invite you to submit a revised version of the manuscript that addresses the points raised during the review process.

I am writing to let you know that your Manuscript ID PONE-D-20-35335 entitled " Health and science-related misinformation on COVID-19. A content analysis of hoaxes identified by fact checkers in Spain" which you submitted to Plos One has now been reviewed.  The comments of the reviewers are included at the bottom of this letter.

Please note that the reviewers have recommended major revisions to your manuscript, which I believe will result in an improved manuscript overall. Particularly, I suggest you attend to clarifying research goals based on a more solid and wide literature review. Therefore, I invite you to revise your manuscript, in conjunction with the comments of the reviewers.

We look forward to receiving your revised manuscript.

Kind regards,

Chang Sup Park, Ph.D.

Academic Editor

PLOS ONE

Additional Editor Comments:

I am writing to let you know that your Manuscript ID PONE-D-20-35335 entitled " Health and science-related misinformation on COVID-19. A content analysis of hoaxes identified by fact checkers in Spain" which you submitted to Plos One has now been reviewed. The comments of the reviewers are included at the bottom of this letter.

Please note that the reviewers have recommended major revisions to your manuscript, which I believe will result in an improved manuscript overall. Particularly, I suggest you attend to clarifying research goals based on a more solid and wide literature review. Therefore, I invite you to revise your manuscript, in conjunction with the comments of the reviewers.

Journal Requirements:

3. Please include additional information regarding the data extraction tool used in the study and ensure that you have provided sufficient details that others could replicate the analyses. For instance, if you developed a data extraction tool as part of this study and it is not under a copyright more restrictive than CC-BY, please include a copy, in both the original language and English, as Supporting Information, or include a citation if it has been published previously.

4. During the internal evaluation of your manuscript. We note that the current article addresses a similar research question with some overlap in the study variables of a  previous publication:  http://profesionaldelainformacion.com/contenidos/2020/may/salaverria-buslon-lopez-leon-lopez-erviti.pdf. In light of the related article, please cite and discuss the above mentioned related paper in the body of the manuscript (introduction, methods if overlap, and discussion). In particular, please discuss how your research contributes to the basis of academic knowledge in light of the above mentioned publication. Please bear in mind that our publication criteria state " If a submitted study replicates or is very similar to previous work, authors must provide a sound scientific rationale for the submitted work and clearly reference and discuss the existing literature. Submissions that replicate or are derivative of existing work will likely be rejected if authors do not provide adequate justification.

Reviewers' comments:

Reviewer's Responses to Questions

**Comments to the Author**

1. Is the manuscript technically sound, and do the data support the conclusions?

Reviewer #1: Partly

2. Has the statistical analysis been performed appropriately and rigorously? 

Reviewer #1: No

3. Have the authors made all data underlying the findings in their manuscript fully available?

Reviewer #1: No

4. Is the manuscript presented in an intelligible fashion and written in standard English?

Reviewer #1: Yes

5. Review Comments to the Author

Reviewer #1: The paper proposes a classification schema of articles evaluated by 3 Spanish fact checking organizations, dubbed "hoaxes" by the authors. The three-month period considered encompasses the first COVID-19 related lockdowns. The authors annotate the hoaxes for kind of media used, platform it appeared on, geographical scope, source type, and a hierarchical "type" classification. Out of 533 articles, authors consider 187 which are about "science and health", and for these, the paper provides tables of class memberships for the above annotations. The authors also provide content analysis, grouping the topics into "origin of the virus", "treatments", "vaccines", etc. They then present analysis using intersection between some of these variables, such as type of hoax vs. format, and type of hoax vs. geographical scope.

The paper suffers from a vague motivation, from the introduction, to the presentation of results, and conclusions. The Abstract, for instance, presents the results in "The "prototypical hoax"..." sentence as a hodge-podge of observations that are difficult to contextualize, and understand their importance. In the Introduction, the authors claim that the "characteristics of these hoaxes and the forms they take have not yet been brought to light", such as [1] concerning India, and I'm sure many more. Similarly, the related work section does not make a compelling case for performing this study. My point is not that observational studies are not important, but they need to be motivated by more than "nobody has done this before". If the classification system proposed here was aimed at a particular task, such as ways to intersect or detect misinformation, or particular theories about communication, the results may be easier to interpret and contextualize.

I commend the authors on coming up with a coding schema that resulted in a higher inter-annotator agreement, giving me confidence that the labels applied in this study are somehow valid.

The analysis presents some interesting insights. For instance, the fact that these hoaxes are often based on "decontextualization" and that this often happens around healthcare/scientific sources. In fact, the authors state several times that the pace of scientific publishing picked up so much that it's likely that even the publishers "were not able to process them properly". The authors provide several examples of poor-quality and retracted articles that made a splash on social media. It would be very interesting to see a systematic analysis of whether pre-publishing articles resulted in retraction and misinformation, instead of seeing this anecdotal evidence.

The fact that lots of these hoaxes were found on social media channels is emphasized by the authors as a major finding in the paper. However, this is largely affected by the article selection policy of the fact-checking websites used in this study. I am not sure about the Spanish mass media, but in the US (with which I am more familiar), plenty of falsehoods are promoted on the television and radio -- media which still have a huge audience, possibly in demographics only somewhat overlapping with that captured by the current study. I would recommend the authors contextualize their findings in this methodological limitation.

Another limitation is the paper's scope - Spain - making the insights highly localized. This is, of course, also a strength, providing points of view outside US-centric research that is so popular. However, the authors fail to really juxtapose their findings to those in other countries, or strongly contextualize the findings in the peculiarities of the Spanish COVID situation at the time. This is another area where the authors could strengthen the paper. For instance, a time visualization could provide readers insights into when most hoaxes happened, and with what major events they were associated. How the political situation may have differed from that of other countries, or influenced by outside political forces (such as Trump's push for hydroxychloroquine). From where the sources of information came (inside or outside Spain), etc.

The authors need to show statistical testing results when claiming comparison, such as on line 341, especially since the final number of documents examined (n) can shrink quite a bit. In the end, this is a sample of all potential misinformation out there -- if we are comparing, say, 5 to 6 documents, is the strength of this difference enough to claim it can be generalized?

Overall, the paper may present a few interesting insights, but is definitely not groundbreaking. It could be strengthened by motivating the labeling exercise by a particular task (say, automated hoax detection, or science decontextualization monitor), a particular theory (ex: pre-publishing scientific news results in misinformation), or emphasis on the peculiarity of the Spanish situation and its interaction with the ongoing events there and abroad. I believe considering the "why is this interesting" question by the authors may make this observational study much more interesting.

Small remarks:

Abstract - "importance of time" should be "importance of timing"?

[1] Akbar, Syeda Zainab, et al. "Misinformation as a Window into Prejudice: COVID-19 and the Information Environment in India." Proceedings of the ACM on Human-Computer Interaction 4.CSCW3 (2021): 1-28.

6. PLOS authors have the option to publish the peer review history of their article (what does this mean?). If published, this will include your full peer review and any attached files.

Reviewer #1: **Yes: **Yelena Aleksandrovna Mejova

---

## [Author Response · Author response to Decision Letter 0]

2 Jun 2021

The responses to reviewers are included in the attached document "Response to reviewers"

---

## [Decision Letter · Decision Letter 1]

19 Oct 2021

PONE-D-20-35335R1

Health and science-related misinformation on COVID-19. A content analysis of hoaxes identified by fact checkers in Spain

PLOS ONE

Dear Dr. León,

Thank you for submitting your manuscript to PLOS ONE. After careful consideration, we feel that it has merit but does not fully meet PLOS ONE’s publication criteria as it currently stands. Therefore, we invite you to submit a revised version of the manuscript that addresses the points raised during the review process.

The reviewer points out that the statistical analysis has not been performed rigorously. Also please make clear conceptual distinctions among hoaxes, rumors, misinformation and fake news.

I look forward to receiving your revision.

We look forward to receiving your revised manuscript.

Kind regards,

Chang Sup Park, Ph.D.

Academic Editor

PLOS ONE

Journal Requirements:

Additional Editor Comments (if provided):

The reviewer points out that the statistical analysis has not been performed rigorously. Also please make clear conceptual distinctions among hoaxes, rumors, misinformation and fake news.

I look forward to receiving your revision.

Reviewers' comments:

Reviewer's Responses to Questions

**Comments to the Author**

1. If the authors have adequately addressed your comments raised in a previous round of review and you feel that this manuscript is now acceptable for publication, you may indicate that here to bypass the “Comments to the Author” section, enter your conflict of interest statement in the “Confidential to Editor” section, and submit your "Accept" recommendation.

Reviewer #2: All comments have been addressed

2. Is the manuscript technically sound, and do the data support the conclusions?

Reviewer #2: Yes

3. Has the statistical analysis been performed appropriately and rigorously? 

Reviewer #2: No

4. Have the authors made all data underlying the findings in their manuscript fully available?

Reviewer #2: Yes

5. Is the manuscript presented in an intelligible fashion and written in standard English?

Reviewer #2: Yes

6. Review Comments to the Author

Reviewer #2: This is an interesting paper on Spanish hoaxes on COVID-19, and I enjoy reading the result. There are some comments or suggestions for further improvement:

Overall comments:

1. The authors have tried to addressed all comments raised in a previous round of review .

the statistical analysis could be improved.

The result is descriptive and lacks knowledge discovery, it would be really nice if the author could consider more comparison or in-depth analysis.

Specific comments：

The statistical analysis has not been performed appropriately and rigorously. For example, for chi-square, which should be used to compare two variable, seems to be used to compare multiple variables at one time (or haven’t been reported properly) as shown in one example below (line 451-453), and p value should not be reported as “p-tail<0,05”:"A significant difference between formats was that audio clips were used more frequently in the case of exaggerations (20.0%) (Table 8) (chi-square= 45,494**; df=30 ; p-tail<0,05). "

The authors use content analysis with Codebook by two coders, there are 3 topics however, in result, only four types are listed, which doesn't match: Healthcare/science,Healthcare/science,Member of the public ,Business.（Table 11. Non-anonymous source types by hoax type.)

I would suggest this research to distinguish the difference between hoaxes, rumors, misinformation and fake news. There have been many studies on misinformation, rumors and fake news, so why choose hoaxes only? Comparison between hoaxes and misinformation and further analysis could be added to enhance the importance of the article.

In the introduction, the author claim to" establish a classification system that can be used to explain the narrative mechanisms that underpin the credibility of this information "(line. 58,59), however, instead of building up a system, this paper demostrates a descriptive analysis on one case only.

7. PLOS authors have the option to publish the peer review history of their article (what does this mean?). If published, this will include your full peer review and any attached files.

Reviewer #2: No

---

## [Author Response · Author response to Decision Letter 1]

23 Nov 2021

-The result is descriptive and lacks knowledge discovery, it would be really nice if the author could consider more comparison or in-depth analysis.

Our results provide knowledge that goes beyond the specific case study. As we explain the conclusion, we have

1. Identified the most common characteristics of health and science related hoaxes.

These characteristics provide relevant information that can be used as a basis for future research. They can also contribute to a better understanding of how disinformation is spread to the public and, therefore, can help to improve media literacy actions about health and science (p. 32).

2. Formulated a typology of this type of hoaxes, according to their relationship to scientific information.

This typology can work as a basis for future research and can help to develop systems for automated detection of health and science- related hoaxes (p. 34).

-The statistical analysis has not been performed appropriately and rigorously. For example, for chi-square, which should be used to compare two variable, seems to be used to compare multiple variables at one time (or haven’t been reported properly) as shown in one example below (line 451-453), and p value should not be reported as “p-tail<0,05”:"A significant difference between formats was that audio clips were used more frequently in the case of exaggerations (20.0%) (Table 8) (chi-square= 45,494**; df=30 ; p-tail<0,05).

The chi-square test was used to compare two variables but the results were not reported properly. We have now amended this in the main text and Appendix 2.

-The authors use content analysis with Codebook by two coders, there are 3 topics however, in result, only four types are listed, which doesn't match: Healthcare/science,Healthcare/science,Member of the public, Business.（Table 11. Non-anonymous source types by hoax type.).

We have amended the manuscript in order to differentiate correctly the four types of hoaxes (deception, decontextualization, exaggeration, and joke/parody.) and the three topics of Science and health-related hoaxes (Scientific research, Scientific policy and health management, Erroneous advice issued to the public): 

The classification of hoax content revealed relatively similar frequencies among the three main topics (p. 17). 

Healthcare/scientific sources predominated in three of the four hoax types (p. 24).

- I would suggest this research to distinguish the difference between hoaxes, rumors, misinformation and fake news. There have been many studies on misinformation, rumors and fake news, so why choose hoaxes only? Comparison between hoaxes and misinformation and further analysis could be added to enhance the importance of the article.

We have clarified this in the text. As the manuscript indicates, the conceptualization regarding the forms of falsehood in publicly disseminated information is based on a “theoretical distinction between disinformation and misinformation”. The first concept refers to deliberate deception, while the second covers inadvertent falsehoods. Ultimately, these two categories distinguish between lying (voluntary) and error (involuntary).

Within these two general categories there are multiple concrete expressions. Specifically, research has explored modalities such as conspiracy theories (Craft et al. 2017), rumors (Alkhodair, 2020) and hoaxes (Braun & Eklund, 2019). In the field of journalism and the media, the so-called “fake” (Tandoc et al., 2021) or “false” (Andı & Akesson, 2020) news have also been widely investigated.

In our study, we have opted for using the concept of “hoaxes”, because the falsehoods investigated do not correspond only to content disseminated in news media and because, according to the existing literature, it is a concept that designates deliberate falsehoods targeted to the general public through any communication channel.

References

Alkhodair, S. A., Ding, S. H., Fung, B. C., & Liu, J. (2020). Detecting breaking news rumors of emerging topics in social media. Information Processing & Management, 57(2), 102018.

Andı, S., & Akesson, J. (2020). Nudging Away False News: Evidence from a Social Norms Experiment. Digital Journalism, 9(1), 106-125.

Braun, J. A., & Eklund, J. L. (2019). Fake news, real money: Ad tech platforms, profit-driven hoaxes, and the business of journalism. Digital Journalism, 7(1), 1-21.

Craft, S., Ashley, S., & Maksl, A. (2017). News media literacy and conspiracy theory endorsement. Communication and the Public, 2(4), 388-401.

Tandoc Jr, E. C., Thomas, R. J., & Bishop, L. (2021). What is (fake) news? Analyzing news values (and more) in fake stories. Media and Communication, 9(1), 110-119.

-In the introduction, the author claim to" establish a classification system that can be used to explain the narrative mechanisms that underpin the credibility of this information "(line. 58,59), however, instead of building up a system, this paper demostrates a descriptive analysis on one case only.

We have amended this phrase and clarified the research objectives in the introduction (p. 5): 

The overall objective of this study is to analyze the science and health related hoaxes about COVID-19 that were spread during the pandemic. More specifically we aim to (1) identify the characteristics of form and content and the platforms used to spread science and health related hoaxes; and (2) formulate a typology that can be used to classify the different types of hoaxes, according to its connection with scientific information.

---

## [Decision Letter · Decision Letter 2]

7 Feb 2022

PONE-D-20-35335R2Health and science-related misinformation on COVID-19. A content analysis of hoaxes identified by fact checkers in SpainPLOS ONE

Dear Dr. León,

Thank you for submitting your manuscript to PLOS ONE. After careful consideration, we feel that it has merit but does not fully meet PLOS ONE’s publication criteria as it currently stands. Therefore, we invite you to submit a revised version of the manuscript that addresses the points raised during the review process.

Please conduct an extensive copy-editing and fix all the statistical errors based on the comments by the reviewer. 

We look forward to receiving your revised manuscript.

Kind regards,

Chang Sup Park, Ph.D.

Academic Editor

PLOS ONE

Journal Requirements:

Reviewers' comments:

Reviewer's Responses to Questions

**Comments to the Author**

1. If the authors have adequately addressed your comments raised in a previous round of review and you feel that this manuscript is now acceptable for publication, you may indicate that here to bypass the “Comments to the Author” section, enter your conflict of interest statement in the “Confidential to Editor” section, and submit your "Accept" recommendation.

Reviewer #3: (No Response)

2. Is the manuscript technically sound, and do the data support the conclusions?

Reviewer #3: Partly

3. Has the statistical analysis been performed appropriately and rigorously? 

Reviewer #3: No

4. Have the authors made all data underlying the findings in their manuscript fully available?

Reviewer #3: Yes

5. Is the manuscript presented in an intelligible fashion and written in standard English?

Reviewer #3: No

6. Review Comments to the Author

Reviewer #3: Work is required on copy-editing to bring the manuscript to publishable standard.

A number of minor statistical errors need to be corrected.

Please refer to detailed comments and suggestions in the Attached document. Thank you.

7. PLOS authors have the option to publish the peer review history of their article (what does this mean?). If published, this will include your full peer review and any attached files.

Reviewer #3: No

---

## [Author Response · Author response to Decision Letter 2]

2 Mar 2022

Responses to reviewers have been included in the document "Response to reviewers", previously uploaded to the system.

---

## [Decision Letter · Decision Letter 3]

7 Mar 2022

PONE-D-20-35335R3Health and science-related disinformation on COVID-19. A content analysis of hoaxes identified by fact checkers in SpainPLOS ONE

Dear Dr. León,

Thank you for submitting your manuscript to PLOS ONE. After careful consideration, we feel that it has merit but does not fully meet PLOS ONE’s publication criteria as it currently stands. Therefore, we invite you to submit a revised version of the manuscript that addresses the points raised during the review process.

The reviewer is satisfied with your work, but suggests some copy editing.

We look forward to receiving your revised manuscript.

Kind regards,

Chang Sup Park, Ph.D.

Academic Editor

PLOS ONE

Additional Editor Comments:

The reviewer is satisfied with your work, but suggests some copy editing.

Reviewers' comments:

Reviewer's Responses to Questions

**Comments to the Author**

1. If the authors have adequately addressed your comments raised in a previous round of review and you feel that this manuscript is now acceptable for publication, you may indicate that here to bypass the “Comments to the Author” section, enter your conflict of interest statement in the “Confidential to Editor” section, and submit your "Accept" recommendation.

Reviewer #3: (No Response)

2. Is the manuscript technically sound, and do the data support the conclusions?

Reviewer #3: Yes

3. Has the statistical analysis been performed appropriately and rigorously? 

Reviewer #3: Yes

4. Have the authors made all data underlying the findings in their manuscript fully available?

Reviewer #3: Yes

5. Is the manuscript presented in an intelligible fashion and written in standard English?

Reviewer #3: No

6. Review Comments to the Author

Reviewer #3: PONE-D-20-35335R2

Health and science-related misinformation on COVID-19. A content analysis of hoaxes identified by fact checkers in Spain

REVIEWER’S COMMENTS

General Comments

Content

The authors are to be commended and thanked for addressing the suggestions and recommendations. The manuscript reads very well. I have no further comments regarding the content.

Well done to the authors on undertaking their thorough research and on preparing a very comprehensive, well-informed and timely manuscript on this important topic. The published article will be of interest to readers around the world!

Copy editing

The manuscript is ready for publication, following minor copy editing (points below) to be undertaken by the authors.

1. p.6, line 148: Delete the comma after ‘… COVID-19’

2. p.7, line 168: Add the word ‘current’, i.e. to read ‘The current research …’

3. p.8, line 188: Add the word ‘period’ immediately before ‘(March 11 to June …)

4. p.10, line 209: Split last word into two, i.e. to read ‘… intercoder reliability’

5. p.10, line 209: I don’t think there is a need for the word ‘and’ between ‘double’ and ‘blind’ as this is usually written simply as ‘double blind’

6. p.11, line 324: Remove comma and second full stop after ‘prominent’

7. Table 6: Place the label (descriptor) on the line above the table, consistent with all other tables.

8. p.19, lines 359-60: After the colon, replace the commas with semi-colons, i.e. to read ‘real; …. ; …. ‘ and …’

9. p.23, line 454: Use a capital ‘T’ for the correct title of The Lancet.

All Tables containing numbers:

10. Line up the ones under ones, tens under tens and hundreds under hundreds in all columns; and

11. Remove brackets in the footnotes to the tables, i.e. remove brackets surrounding * and **.

Tables containing words:

12. Adjust either width of some columns or size of font to ensure that words are not split incorrectly (e.g. across two lines) and that commas appear immediately after the relevant word.

7. PLOS authors have the option to publish the peer review history of their article (what does this mean?). If published, this will include your full peer review and any attached files.

Reviewer #3: No

---

## [Author Response · Author response to Decision Letter 3]

8 Mar 2022

Response to reviewer’s comments

1. p.6, line 148: Delete the comma after ‘… COVID-19’. 

Done.

2. p.7, line 168: Add the word ‘current’, i.e. to read ‘The current research …’

Done.

3. p.8, line 188: Add the word ‘period’ immediately before ‘(March 11 to June …)

Done

4. p.10, line 209: Split last word into two, i.e. to read ‘… intercoder reliability’

Done.

5. p.10, line 209: I don’t think there is a need for the word ‘and’ between ‘double’ and ‘blind’ as this is usually written simply as ‘double blind’

Done.

6. p.11, line 324: Remove comma and second full stop after ‘prominent’

Done.

7. Table 6: Place the label (descriptor) on the line above the table, consistent with all other tables.

Done.

8. p.19, lines 359-60: After the colon, replace the commas with semi-colons, i.e. to read ‘real; …. ; …. ‘ and …’

Done.

9. p.23, line 454: Use a capital ‘T’ for the correct title of The Lancet.

Done.

All Tables containing numbers:

10. Line up the ones under ones, tens under tens and hundreds under hundreds in all columns; and

Done.

11. Remove brackets in the footnotes to the tables, i.e. remove brackets surrounding * and **.

Done.

Tables containing words:

12. Adjust either width of some columns or size of font to ensure that words are not split incorrectly (e.g. across two lines) and that commas appear immediately after the relevant word. 

Done.

---

## [Editor Report · Decision Letter 4]

14 Mar 2022

Health and science-related disinformation on COVID-19. A content analysis of hoaxes identified by fact checkers in Spain

PONE-D-20-35335R4

Dear Dr. León,

We’re pleased to inform you that your manuscript has been judged scientifically suitable for publication and will be formally accepted for publication once it meets all outstanding technical requirements.

Kind regards,

Chang Sup Park, Ph.D.

Academic Editor

PLOS ONE
---

## [Editor Report · Acceptance letter]

18 Mar 2022

PONE-D-20-35335R4 

Health and science-related disinformation on COVID-19:
A content analysis of hoaxes identified by fact-checkers in Spain 

Dear Dr. León:

I'm pleased to inform you that your manuscript has been deemed suitable for publication in PLOS ONE. Congratulations! Your manuscript is now with our production department. 

Kind regards, 

on behalf of

Dr. Chang Sup Park 

Academic Editor

PLOS ONE